# Intelligent Predicting of Product Quality of Injection Molding Recycled Materials Based on Tie-Bar Elongation

**DOI:** 10.3390/polym14040679

**Published:** 2022-02-10

**Authors:** Hanjui Chang, Zhiming Su, Shuzhou Lu, Guangyi Zhang

**Affiliations:** 1Department of Mechanical Engineering, College of Engineering, Shantou University, Shantou 515063, China; 19zmsu@stu.edu.cn (Z.S.); 21szlu@stu.edu.cn (S.L.); 20gyzhang1@stu.edu.cn (G.Z.); 2Intelligent Manufacturing Key Laboratory of Ministry of Education, Shantou University, Shantou 515063, China

**Keywords:** recycled material, molding quality, clamping force, tie-bar elongation, injection parameters

## Abstract

In the process of injection molding, a certain percentage of recycled material is usually used in order to save costs. The material properties of recycled materials can change significantly compared with raw materials, and the quality of their molded products is more difficult to control. Therefore, it is crucial to propose a method that can effectively maintain the yield of the recycled material products. In addition, the variation of clamping force during the injection molding process can be determined by measuring the tie-bar elongation of the injection molding machine. Therefore, this study proposes a real-time product quality monitoring system based on the variation of clamping force during the injection molding process for the injection molding of recycled materials for plastic bottle caps. The variation of clamping force reflects the variation of cavity pressure during the injection molding process and further maps the variation of injection parameters during the injection molding process. Therefore, this study evaluates the reliability of the proposed method for three different injection parameters (residual position, metering end point and metering time). Experiments have shown that there is a strong correlation between the quality (geometric properties) and weight of the product under different molding parameters. Moreover, the three main injection parameters have a strong influence on the weight and quality of the plastic caps. The variation of the clamping force is also highly correlated with the weight of the plastic bottle cap. This demonstrates the feasibility of applying the variation of clamping force to monitor the quality of injection molded products. Furthermore, by integrating the clamping force variation index with the calibration model of the corresponding injection parameters, it is possible to control the weight of the plastic cap within the acceptable range of the product in successive production runs.

## 1. Introduction

Injection molding is a form of mass plastic product manufacturing, which has the advantages of low cost, high efficiency, and the ability to produce precision parts, and it is widely used in the fields of electronic products, automobiles, medical devices and optical lenses. We can say that plastic products are everywhere in our life. However, in the process of injection molding, plastic waste or waste plastic products will inevitably be produced, and plastic may take hundreds of years to degrade in the natural environment, so for reasons of environmental protection and cost saving, the production process of injection molding usually occurs using recycled materials. After the original material is molten and compressed and cooled into recycled material, its flow characteristics with the raw material change significantly, affecting the quality of the injection molded product, so it is crucial to propose a method that can effectively maintain the yield of the product with recycled material. Theoretically, the current high-precision injection molding machines can automatically meet the requirements of high precision and high yield of injection molded products. However, in the actual production process, the use of recycled materials can lead to unstable quality changes of molten plastic, resulting in unstable injection molding product quality and failure to ensure high product yield production. This is mainly because the high-precision injection molding machine only ensures the stability of parameters during the molding process and does not control the unstable changes of molten plastic quality. Moreover, most manufacturers currently check the quality of injection molded products by means of off-line sampling to check the geometry and indicated quality of products, which is a time-consuming and costly inspection method. Therefore, proposing a method that enables real-time online monitoring, control and prediction of the molding quality of injection molded products made of recycled materials is a worthy research topic.

## 2. Literature Review

The quality of injection molded products is closely related to the material properties of the melt (e.g., flowability, etc.) and because the injection molding process is complex, it is influenced by many different factors. In particular, when using recycled materials for injection molding, it is difficult to maintain the consistency of product quality because the material properties between recycled materials and virgin materials are different (i.e., viscosity, shrinkage and fluidity), and this becomes more difficult as the differences increase [1]. Therefore, when using recycled materials for injection molding, it is worth considering how to monitor and control the impact of changes in material properties on product quality and thus maintain high product yields. In order to improve the quality of recycled material products, in 2016, GU et al. [2] analyzed and evaluated recycled polypropylene (PP) composites with talc as the main filler and maleic anhydride-grafted polypropylene (MAPP) as the bulking agent, all results were analyzed using principal component analysis (PCA) and cluster analysis (CA); the results show that adding fillers can well improve the physical and mechanical properties of virgin PP composite and its recycled plastic mechanical properties. In 2017, Latiff [3] investigated the effect of adding different fiber conditions, fiber loading, and chemically modified carbon fibers to polypropylene (PP) on the physical, mechanical and frictional properties of PP, and the results show that carbon fibers with certain suitable conditions had a significant effect on improving the physical and mechanical properties of PP materials, etc. In 2019, Yu et al. [4] showed the effect of adding different loading levels of lignin to recycled polypropylene (PP) composites on their related mechanical and physical properties, and the results show that the addition of lignin in this study was able to successfully improve the mechanical properties of recycled PP composites and their oxidation resistance.

It can be seen that the above researchers only added the appropriate additives or fibers to the recycled material to enhance the mechanical properties of the recycled material. However, as more additives are added to the recycled material, it leads to an increase in the difference between it and the original material, making the stability of the product quality less likely to be guaranteed. Generally speaking, the quality of injection molded products is inseparably related to the material properties of the melt. In addition, in the injection molding process of recycled materials, the material properties of their melts are affected by temperature and pressure.

In 2019, Barghikar et al. [5] experimentally monitored the variation of pressure and temperature inside the injection mold and compared them with simulation results to investigate their effect on lens warpage deformation. This study shows that the variation of pressure and temperature inside the cavity is one of the important factors affecting the warpage of the lens and its geometric quality, which can have an effect on the warpage of the spherical lens, and the amount of warpage of the lens is smaller when the slope of the pressure curve inside the mold cavity is lower. In 2019, Macedo et al. [6] investigated the effect of dynamic temperature control on the injection molding process based on dynamic mold heating and cooling control techniques for rapid heating and cooling molding (RHCM) and improved the efficiency and product yield of variable temperature molds when applied to injection molding manufacturing under certain controlled conditions. In 2020, Chen et al. [7] developed an artificial neural-network-based online defect detection system for injection molded products by extracting the temperature and pressure signals in the cavity, and this study demonstrates that the monitoring of temperature and pressure in the cavity during the injection molding process by this system can achieve stable control of the injection molding product quality and improve the product yield. In 2021, Karagöz [8] studied the effect of mold surface temperature on injection molded products in HDPE injection molding and prepared experimental samples by varying the mold’s surface temperature. Therefore, we can monitor the temperature or pressure signal during the injection molding process to adjust and control the material characteristics of the melt during the injection molding process to ensure the stability of product quality. However, in 2016, Rusdi et al. [9] applied computer-aided techniques for numerical simulation analysis to investigate the effect of two working parameters, pressure and temperature, on the injection molding process. In addition, by conducting actual injection molding and rheological experiments, the results show that system pressure dominates the influence on key injection molding process parameters such as filling time and speed, while temperature can only vaguely affect the current ground injection molding process. 

In other words, for the injection molding process, the influence of system pressure variation on the injection molding process is more significant than the influence of temperature variation on the injection molding process. In general, the in-mold temperature is monitored by drilling holes directly into the mold and installing embedded temperature sensors, but this monitoring method is difficult due to the obstruction of some complex in-mold structures (complex waterways, model structures, etc.). Therefore, in this study, we mainly discuss the purpose of controlling the product quality by controlling the relevant pressure parameters in the injection molding process.

During the injection molding process, the variation of pressure in the mold cavity is critical to the quality of the molded product, and it can be said that the cavity pressure is the dominant factor in determining the quality of the product. In other words, the control of cavity pressure can be an important means to improve the repeatability and consistency of product quality in the injection molding process. If real-time and effective monitoring and adjustment of cavity pressure signals can be achieved, then the quality of injection molded products, including those produced by applying recycled materials, can also be effectively improved. In 2014, Reiter et al. [10] proposed a method for predicting the cavity pressure controller using a model based on a physically motivated gray box model. The model can visually check the rationality of the cavity pressure controller and can reproduce the dominant behavior of the system. This is effective in improving the repeatability and consistency of cavity pressure and product molding quality for each injection molding process. In 2016, Nam et al. [11] developed a diagnostic and error prediction model for lens injection molding based on the response surface method to extract the signal of cavity pressure during the injection molding process and measure the shape error of the lens they want to produce by embedding a pressure sensor in the mold to improve the yield of the lens in actual injection molding. In 2017, Hopmann et al. [12] applied an iterative learning control method to optimize the control strategy of cavity pressure to improve the injection molding machine accuracy and further improve the injection molding product quality based on the PVT (pressure, volume and temperature) characteristics of the material, i.e., considering the relationship between pressure, volume and temperature at the same time and using the characteristic that the injection molding process is repetitive. In 2020, Stemmler et al. [13] proposed the concept of model-based self-optimization in injection molding and applied a model-based parametric optimization iterative learning controller (NOILC) to monitor the variation of cavity pressure during injection molding, and the experiments of this study show that their proposed method could achieve high-precision control of cavity pressure and improve the product quality, especially the weight stability. In 2021, Huang et al. [14] proposed a new method to regulate and control holding pressure during the injection molding process to control and regulate the cavity pressure and reduce its deviation in each injection molding country and verified the feasibility of this research method to improve the consistency of the injection molding product quality by performing injection molding experiments on thin-walled dumbbell-shaped samples. In addition, in 2021, Huang et al. [15] applied a calibration model generated by machine learning to improve the effectiveness of numerical simulation of polymer melt flow behavior in the cavity for predicting and optimizing injection molding process parameters, especially to improve the accuracy of simulation results of cavity pressure profiles during injection molding, providing an effective reference for determining the settings of molding parameters related to cavity pressure. In 2021, Chen et al. [16] proposed a least-squares regression-based method using the characteristics of cavity pressure during molding to determine the appropriate V/P switching point and related molding parameters according to the dimensional requirements of the product. In addition, according to setting a reasonable multi-stage pressure-holding method, the residual pressure difference between the near-gate and far-gate areas of the mold at the end of pressure-holding is reduced to improve the geometric uniformity of injection molded products.

All of the above studies were conducted to improve the yield of injection molded products by directly extracting the pressure signal inside the cavity and controlling the regulation. In 2019, Poszwa et al. [17] proposed that in the injection molding process, clamping force is needed to keep the injection molding machine mold in the closed state, so the change in pressure in the cavity causes a corresponding change in clamping force in the injection molding machine. Therefore, based on the relationship between cavity pressure and clamping force changes in the injection molding process, we can extract the real-time clamping force change signal in the parallel molding process, which can be used to indirectly reflect the change in cavity pressure in the process. In 2019, Chen et al. [18] monitored and controlled the change in clamping force of the injection molding machine in real time by installing strain sensors directly on the tie bars of the injection molding machine and successfully reflected the change in pressure in the cavity as the change in clamping force, which enabled the part quality to be controlled within a small error range. In 2019, Zhang et al. [19] achieved indirect monitoring of cavity pressure by using ultrasonic technology to evaluate the stress on the tie bars of the injection molding machine and investigated the effect of clamping force on the measured cavity pressure profile; the results of the study show that selecting the appropriate clamping force could obtain more accurate measurement results and achieve real-time nondestructive monitoring of cavity pressure during the injection molding process. In 2013, Zhao et al. [20] proposed this method to indirectly reflect and control the change in cavity pressure during the injection molding process, which can ensure the nondestructive measurement of cavity pressure during the injection molding process and improve the accuracy of cavity pressure for monitoring, optimizing and controlling the injection molding process compared with the traditional way of drilling holes directly on the mold and monitoring the cavity pressure through embedded pressure sensors. This improves the accuracy of cavity pressure for monitoring, optimizing, and controlling the injection molding process.

At the same time, there is a close relationship between the variation of cavity pressure and product weight during injection molding, and generally speaking, the higher the cavity pressure, the heavier the product weight will be. In addition, in 2013, Hassan [21] studied the effect of parameters, especially cavity pressure, on the weight of final injection molded products in injection molding, and the experimental results show that under certain conditions of control, the cavity pressure decreased while reducing the product weight, reflecting the corresponding relationship between cavity pressure and product weight. So, there is also a certain correlation between the change in clamping force of injection molding machine and product weight. In addition, the injection parameters of the residual position, metering end point and metering time are closely related to the final amount of melt injected into the cavity, i.e., these three injection parameters affect the weight of the final molded product, so there is a correlation between the clamping force of the injection molding machine and them. Based on this, to address the problem of inconsistent molding quality of plastic bottle caps, and based on the variation of the clamping force of the plastic machine during injection molding and its correlation with three injection parameters (residual position, metering end point and metering time), this study proposes a product quality monitoring system to achieve adaptive control of the injection molding process in order to improve the stability of product quality, especially the stability of product quality after the application of recycled materials to the injection molding production of plastic bottle caps.

## 3. Materials and Methods

In the method of quality monitoring of recycled material molded products proposed in this study, the variation of the surface strain of the tie bar is measured by four strain sensors installed on each of the four tie bars of the injection molding machine (as shown in Figure 1). When the tie bar of the injection molding machine is stretched in the axial direction, its radial dimension is reduced. Conversely, when the tie bar of the injection molding machine is compressed in the axial direction, its radial dimension is enlarged. In the process of injection molding, the axial dimension of the tie bar of the injection molding machine is stretched when the mold is closed. Especially, when the molten rubber fills the cavity and requires a large clamping force for compression, the tie bar of the injection molding machine is further stretched. In this case, the strain on the surface of the tie bar of the injection molding machine caused by its radial contraction can be recorded by the strain sensor. The stresses and strains on the surface of the tie bar of the injection molding machine during axial stretching are derived from Hooke’s law as shown in Equations (1) and (2).
(1)$$\sigma_{i}=E\varepsilon_{i}$$
(2)$$\sigma_{i}=\frac{F_{i}}{S}$$
where, $\sigma_{i}$ denotes the stress of the $i^{\mathrm{th}}$. tie bar in MPa; E denotes the Young’s modulus of the tie bar of the injection molding machine in $N/{\mathrm{mm}}^{2}$; $\varepsilon_{i}$ denotes the strain of the $i^{\mathrm{th}}$ tie bar; *S* denotes the cross-sectional area of the tie bar of the injection molding machine in ${\mathrm{mm}}^{2}$ and $F_{i}$ denotes the clamping force generated by the $i^{\mathrm{th}}$ tie bar. From the above equation, the total clamping force F (in kN) of the injection molding machine in this study can be derived, as shown in Equation (3).
(3)$$F=\sum_{i=1}^{n}F_{i}=\sum_{i=1}^{n}ES\varepsilon_{i}$$

During the pressure-holding stage of the injection molding process, the male and female molds usually have a certain separation due to the sudden change in pressure in the cavity, when the separation distance between the male and female molds exceeds a certain tolerable range, it leads to burrs on the injection molded products and even reduces the working life of the injection molding machine. Therefore, it is necessary to control the separation between male and female molds within a reasonable range to avoid burrs and other injection defects and to maintain the normal working life of the injection molding machine. As shown in Figure 2, during the injection molding process, when the filling ends and enters the pressure-holding stage, the melt in the cavity is compressed, resulting in a sudden increase in the pressure in the cavity, which requires a greater clamping force and thus leads to the instantaneous elongation of the tie bar. Generally speaking, the extension value of the tie bar of the injection molding machine is between about 8–12 um for a small deformation (for pure material) and between about 13–27 um for a large deformation (for recycled material). Therefore, in this study, the strain sensors installed on the four tie bars of the injection molding machine are used to monitor the sudden change in strain on the surface of the large column of the injection molding machine, thus reflecting the change in clamping force, and this change in clamping force is used as an index to evaluate the injection quality of the product.

In this study, the feasibility of using the change in clamping force of an injection molding machine as a quality indicator for assessing injection molded products was evaluated by assessing the correlation between the change in clamping force of the injection molding machine (i.e., the quality indicator of the injection molded product) as reflected by the change in strain on the surface of the tie bars captured by the strain sensors mounted on the four tie bars of the injection molding machine and the related injection molded product quality. In this study, the Pearson correlation coefficient (PCC) [22] was used to verify the correlation between the variation of the clamping force of the injection molding machine and the quality of its molded products. Equation (4) is expressed as the equation of Pearson’s correlation coefficient (r), which takes values in the range (−1, 1), where x represents the variation of clamping force of the injection molding machine (quality index) and y represents the quality of the molded product. Table 1 shows the correlation strength corresponding to different values of r. The higher the value of |r|, the stronger the correlation between the quality index of injection molded product and its molding quality.
(4)$$r=\frac{\sum^{\;}\left(x-\bar{x}\right)\left(y-\bar{y}\right)}{\sqrt{\sum^{\;}{\left(x-\bar{x}\right)}^{2}}\sqrt{\sum^{\;}{\left(y-\bar{y}\right)}^{2}}}$$

In the real-time product quality monitoring system proposed in this research, the main purpose is to automatically adjust the residual position based on the clamping force of the injection molding machine as reflected by the extension of the tie bar detected by the strain sensor, in order to maintain the weight of the injection molded product for successive production mold times. For the plastic bottle cap, the injection molded product in this study, the nominal weight is 2.27 g (obtained from the volume of the mold cavity and the density of the melt injected into the cavity), with an acceptable upper and lower deviation of 0.1 g. As shown in Figure 3, the flow chart of the real-time monitoring method of the injection molding product quality proposed in this research is as follows: firstly, the standard setting of the clamping force of each tie bar of the injection molding machine is defined as $F_{i}$, and its upper and lower limits are defined as $F_{i}\pm 2\sigma$ (σ indicates the standard deviation); the actual clamping force $F_{i}{}^{\prime }$ of each tie bar of the injection molding machine calculated by the strain sensor detection, if $F_{i}{}^{\prime }$ is not within the defined upper and lower limits of the clamping force of each tie bar, the residual position set by the current injection molding machine should be adjusted in a timely manner (that is, when $F_{i}{}^{\prime }>F_{i}+2\sigma$, then the residual position of the injection molding machine should be set in advance to 0.2 mm; when $F_{i}{}^{\prime }<F_{i}+2\sigma$, then the residual position of the injection molding machine should be set delayed to 0.2 mm), and the opposite means that the injection molding machine can work normally; confirm whether the clamping force on each tie bar of an injection molding machine is back to the upper and lower limit defined after adjusting the remaining position. If the clamping force is within the upper and lower limits, the control adjustment logic is stopped and the production is continued with the current adjusted remaining position. When that is not the case, the linear extrapolation model is applied to continue to control and adjust the residual position until the clamping force on each tie bar of the injection molding machine meets the defined upper and lower limits.

During the actual injection molding production process, the product real-time quality monitoring system of this study operates continuously, triggering and executing the above-mentioned adjustment control process when the corresponding clamping forces detected through the four tie rods are not within the defined range of clamping forces. In addition, the standard clamping force $F_{i}$ and its upper and lower limits $F_{i}\pm 2\sigma$ for a single tie bar in the above-mentioned adjustment control process are derived by observing and counting dozens of continuously produced injection molds.

## 4. Case Study

The overall shape of the injection molding machine used in this research experiment is shown in Figure 4 and its detailed specifications are shown in Table 2. Figure 5a,b shows the geometry of the plastic bottle caps in this study and a sample of their finished injection molded products with an average thickness of 2.3 mm, respectively. The material used in the production of the caps was PE and its model number was “PE GHR 8020 Celanese”. In this study, the original, recycled and triple recycled PE materials were used for injection molding experiments, as shown in Table 3. Moreover, the plastic pellets in different states were not mixed in this study.

In addition, the structure of the experimental measurement of plastic bottle cap injection molding in this study is schematically shown in Figure 6, including an injection molding machine, strain sensors, a data acquisition (DAQ) interface and a computer. Among them, four strain sensors ($S_{1},\;S_{2},\;S_{3}\;\mathrm{and}\;S_{4}$) are mounted on the four tie rods of the injection molding machine. This study was conducted to evaluate the molding quality of plastic bottle caps by measuring the geometry of the three critical circumferences of the caps ($\emptyset_{1}=1.69\;\mathrm{mm}$, $\emptyset_{2}=1.20\;\mathrm{mm}$ and R=0.95 mm) shown in Figure 5a and their weights.

According to the above experimental setup, injection molding experiments of plastic bottle caps in this study were conducted to investigate the effects of three main process parameters, namely, margin position (18.5~20.5 mm), metering end point (94.0~95.0 mm) and metering time (7.5~8.5 s), on the quality of injection molded products. Figure 7 shows the weight of the plastic bottle cap and the corresponding geometry of the critical circumference in 30 successive injections of the original PE material. The data of their relevant experimental results are shown in Table 4. The correlation between the weight of the product and the geometry of its corresponding circumference can be calculated by the Pearson correlation coefficient formula shown in Equation (4), where x and y are the weight of the plastic cap and the geometry of the corresponding critical circumference, respectively. Through experimental calculations, the weight of the plastic bottle cap and the corresponding geometry of the critical circumference in this study both show a high correlation (${\left|r\right|}_{\emptyset_{1}}=0.99$, ${\left|r\right|}_{\emptyset_{2}}=0.99$ and ${\left|r\right|}_{R}=0.97$), so it can be confirmed that the weight of the product can indeed be used as a valid index to assess the quality of product molding. It is indeed easier to fit a curve based on three points, but each data in this study was obtained by conducting multiple experiments under the same conditions, excluding a certain degree of chance, so the correlation coefficients derived from this are also of some reference value.

The important parameters related to the injection molding experiments for this study are shown in Table 5. Using the original PE material, quantitative analysis experiments were conducted for the three main process parameters of margin position, metering end point and metering time. The number of combinations of the three process parameters was 9, as shown in Table 6, and 10 experiments were conducted for each experimental combination. As shown in Figure 8, which depicts the changes of the clamping force and the product weight of the experiment with different parameter variations, it can be seen that the product weight decreases with the increase in the margin position or increases with the increase in the metering end point and metering time. The results of their relevant experimental data are shown in Table 7. This is because the larger the margin position, the less melt is injected into the cavity, and less melt leads to smaller cavity pressure, which in turn leads to less clamping force and less product weight. For the metering end point, the larger the metering end point is, the more melt is plasticized in the storage stage and the more melt is injected into the cavity, resulting in larger cavity pressure, so the product weight increases, and the clamping force increases. Finally, for the metering time, a longer metering time also leads to more melts being injected into the cavity, which also causes an increase in clamping force and product weight. The analysis of the experimental results also shows that there is a strong correlation between the change in clamping force and the three main process parameters: margin position, metering end point and metering time, which have Pearson coefficients of ${\left|r\right|}_{\mathrm{margin}\;\mathrm{position}}=0.98$, ${\left|r\right|}_{\mathrm{metering}\;\mathrm{end}\;\mathrm{point}}=0.97$ and ${\left|r\right|}_{\mathrm{metering}\;\mathrm{time}}=0.99$, respectively. This result indicates that the amount of variation of clamping force can be used as a quality index to evaluate the quality of product molding, which provides a strong basis for the subsequent experimental study.

The flow characteristics of primary PE material and recycled material are different, which may affect product quality. Therefore, in this study, we conducted 100 consecutive injection molding tests using the original PE material and its recycled material (once-recycled material and double-recycled material) with and without the application of this research method to verify the feasibility of the product quality monitoring method proposed in this study for monitoring the method for injection molding of recycled materials. In addition, in order to verify the influence of the flow characteristics of different materials on the product quality, the melt temperature was set to 220 °C for the first 50 molds and 230 °C for the last 50 molds of each material. The higher melt temperature decreases the viscosity of the melt (the viscosity curve of the original PE material for this study is shown in Figure 9), and the better the flow ability, the more melt is injected into the cavity, so the weight of the product and the clamping force increase with the increase in the melt temperature. Figure 9 shows the results of the clamping force change and the corresponding product weight obtained from the experiments using the original PE material and the recycled material (once-recycled material and double-recycled material). The red dashed line shows the upper and lower limits of the product weight and the black dashed line shows the upper and lower limits of the clamping force. We can see that when the product quality monitoring method of this study is not applied (Figure 10a,c,e), some of the product weights are still outside the acceptable upper and lower limits, even at a low melt temperature (220 °C); when the product quality monitoring method of this study is applied (Figure 10b,d,f), most of the product weights fall within the acceptable upper and lower limits throughout the experiment. The effect of melt temperature on product quality was controlled at this time.

Table 8 shows the results of the mean, range and standard deviation of the product weights produced using virgin material and once- and double-recycled materials with and without the application of the quality control method of this study, respectively. Among them, the range and standard deviation of the product weights produced by applying the quality control method of this study are lower than those not produced by applying this method, except for the results of the double-recycled material. This is because although the melt quality of the melt glue changes with the increase in temperature, the product weight results can still converge to the qualified range of plastic bottle cap weight ($2.27_{-0.1}^{+0.1}g$) after applying the quality control method of this study. As for the double-recycled material, after multiple recyclings, its melt quality characteristics are more different from the product, so its convergence results are not very good after applying the quality monitoring method in this study. When the original material undergoes the injection molding process, some of its molecular bonds will be broken due to the increase in temperature, and its material properties will be different compared with the original material. However, in general, by applying the quality monitoring system of this study, the influence of the melt quality with the increase in temperature on the product quality can be controlled to meet the product quality requirements. Therefore, the method proposed in this study to control the product quality based on the large column elongation is effective.

## 5. Conclusions

The stability of the quality of injection molded products fluctuates due to changes in the flow characteristics of the melt, and the large differences in material properties between recycled and virgin materials can cause instability in product quality and prevent the production of high yield products. Therefore, an effective method is needed to monitor the melt quality and related process parameters in real time and to achieve adaptive process control. In this study, we propose a method for real-time online monitoring, control and prediction of the molding quality of injection molded products made of recycled materials. The method monitors the elongation of the tie-bars of the injection molding machine by means of strain sensors installed on the four tie-bars of the injection molding machine, and it investigates its influence on the quality of the injection molded products according to its correspondence with the clamping force of the injection molding machine. That is, this study adjusts and calibrates the residual position by the change in the elongation of the tie-bars and thus controls the weight of the plastic bottle cap. The following main conclusions were drawn from the experiments of this study:
(1)Using Pearson’s correlation coefficient, the correlation between the weight of the plastic bottle cap and the corresponding geometric dimensions of the critical circumference in this study was investigated to verify that the product weight can be used as a valid indicator to assess the molding quality of the product. The experimental results indicate that both the weight of the plastic bottle cap and its corresponding geometric dimensions of the critical circumference in this study exhibited high correlation (${\left|r\right|}_{\emptyset_{1}}=0.99$, ${\left|r\right|}_{\emptyset_{2}}=0.99$ and ${\left|r\right|}_{R}=0.97$), confirming that the product weight can be used as a valid indicator to assess the molding quality of the product.(2)Using the original PE material, quantitative analysis experiments were conducted to investigate the effects of three main process parameters, namely the residual position (18.5–20.5 mm), metering end point (94.0–95.0 mm) and metering time (7.5–8.5 s), on the molding quality of plastic bottle caps and to explore the relationship between them and the amount of change in clamping force of the injection molding machine. From the experimental results, it can be concluded that the product weight decreases with the increase in margin position or increases with the increase in metering end point and metering time, and there is a strong correlation between the variation of clamping force and the three main process parameters of margin position, metering end point and metering time; the Pearson coefficients between them are ${\left|r\right|}_{margin\;position}=0.98$, ${\left|r\right|}_{metering\;end\;point}=0.97$ and ${\left|r\right|}_{metering\;time}=0.99$, demonstrating that the amount of variation in clamping force can be used as a reliable quality indicator to assess the molding quality of the product.(3)The feasibility of the product quality monitoring method proposed in this study for injection molding of recycled material was investigated by using the original PE material and its recycled material (once-recycled material and double-recycled material) in two different situations, with and without the application of this study’s method. It can be seen that after the application of this research method, the weight of the plastic bottle cap falls more steadily within the standard tolerance range under the variation of the melt quality caused by the temperature change, which proves the feasibility of this research method for injection molding of recycled material.(4)For both the original material and the recycled material, the experiments in this study show that the quality control method proposed achieved good control of the weight of the plastic bottle cap, which verified the effectiveness of the method proposed in this study, especially for the quality control of the recycled material when it is used in the production of injection molded products.


## Figures and Tables

**Figure 1 polymers-14-00679-f001:**
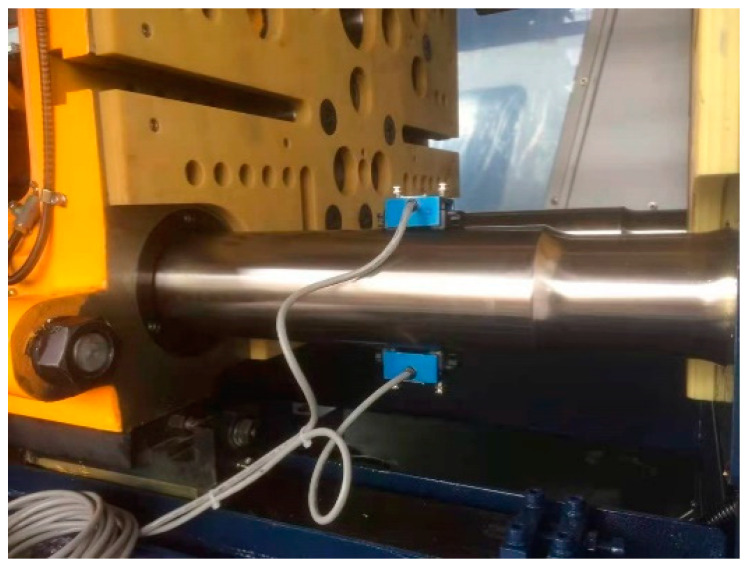
This study’s strain sensor and its position on the tie bar.

**Figure 2 polymers-14-00679-f002:**
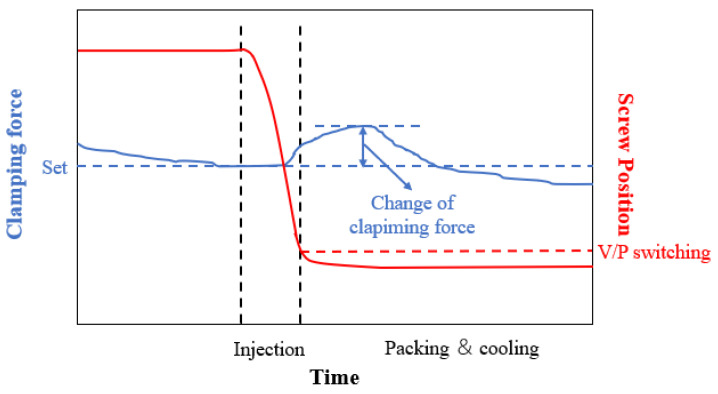
Schematic diagram of clamping force change.

**Figure 3 polymers-14-00679-f003:**
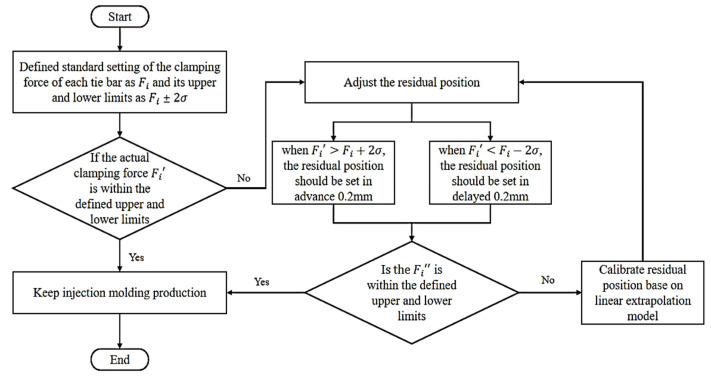
The flow chart of the monitoring method of the injection molding product quality proposed.

**Figure 4 polymers-14-00679-f004:**
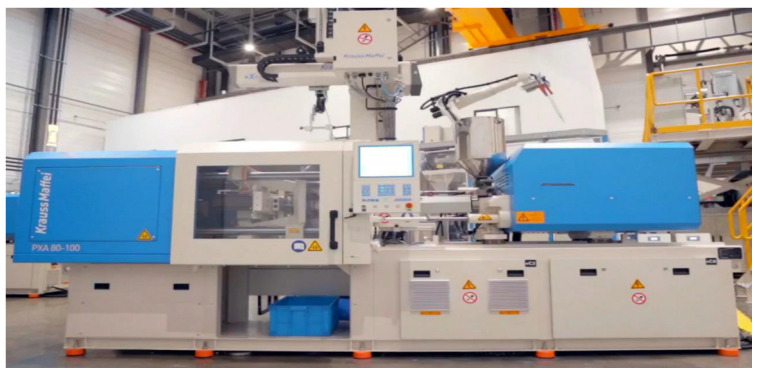
The injection molding machine used in this study’s experiment.

**Figure 5 polymers-14-00679-f005:**
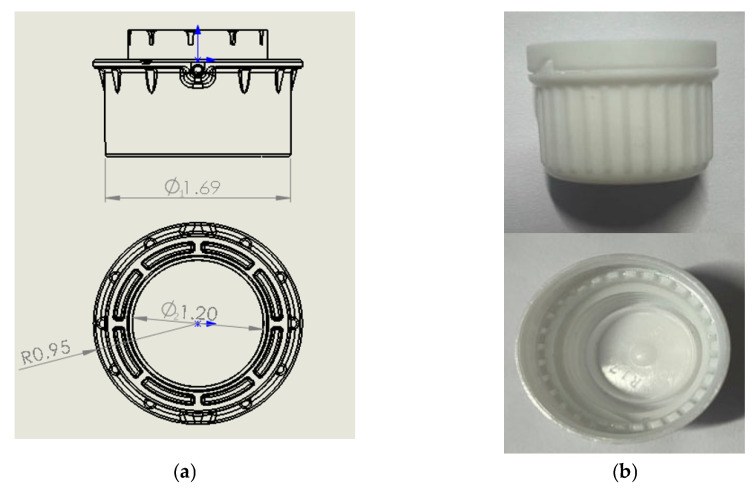
(**a**) Geometric dimensions of plastic bottle caps for this study; (**b**) Sample of plastic bottle cap.

**Figure 6 polymers-14-00679-f006:**
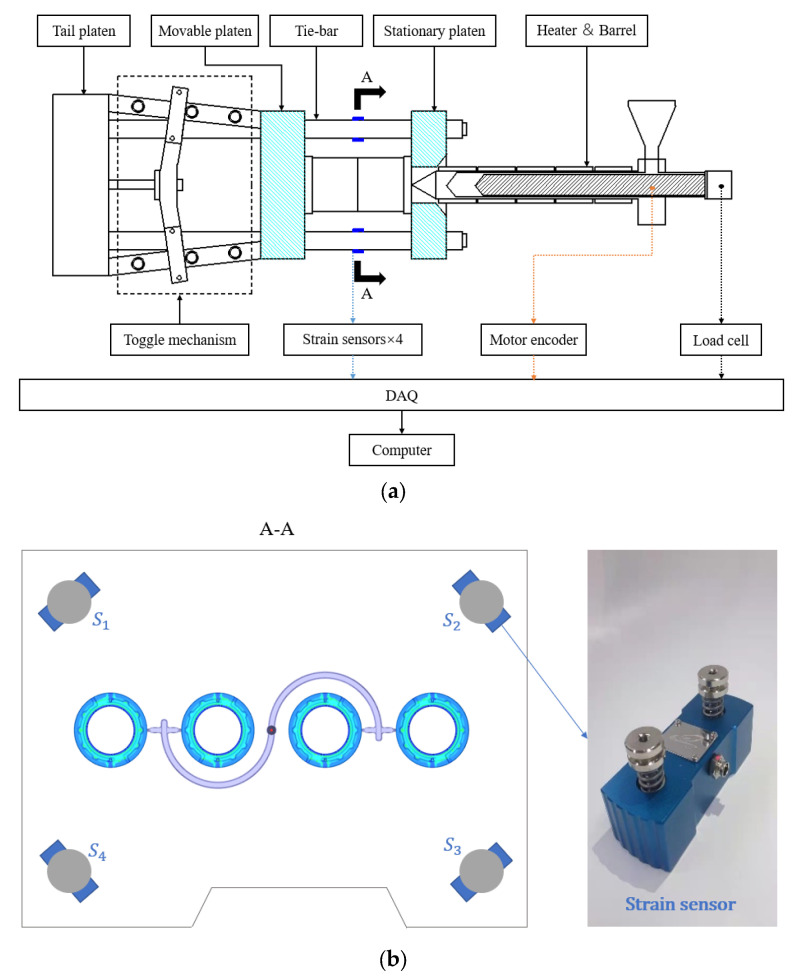
(**a**) Schematic diagram of the injection molding machine and quality monitoring and control system for this study; (**b**) A-A section diagram of injection molding machine.

**Figure 7 polymers-14-00679-f007:**
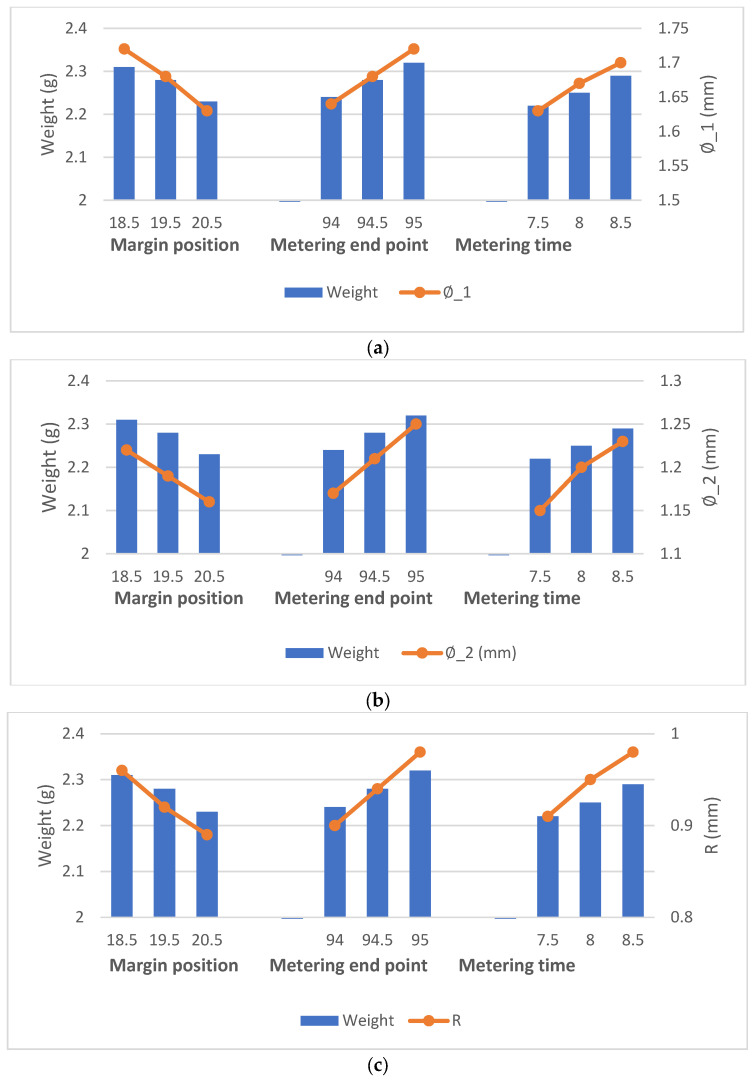
Changes of plastic bottle cap weight and corresponding key circumferential geometric dimensions under different injection molding parameters (margin position, metering end point, metering time). (**a**) Changes of plastic bottle cap weight and $\emptyset_{1}$ under different injection molding parameters; (**b**) Changes of plastic bottle cap weight and $\emptyset_{2}$ under different injection molding parameters; (**c**) Changes of plastic bottle cap weight and *R* under different injection molding parameters.

**Figure 8 polymers-14-00679-f008:**
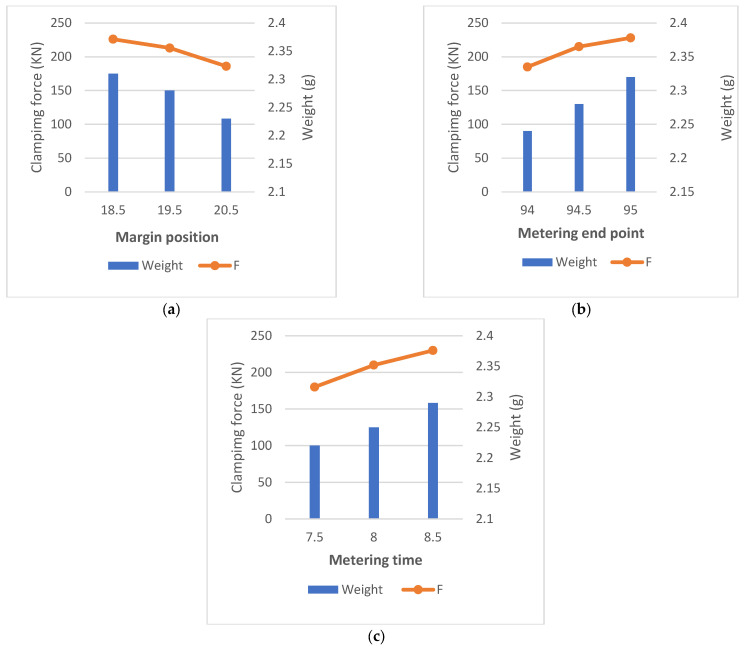
Correlation between clamping force and product weight for different molding parameters. (**a**) Correlation between clamping force and product weight when margin position changes; (**b**) Correlation between clamping force and product weight when metering end point changes; (**c**) Correlation between clamping force and product weight when metering time changes.

**Figure 9 polymers-14-00679-f009:**
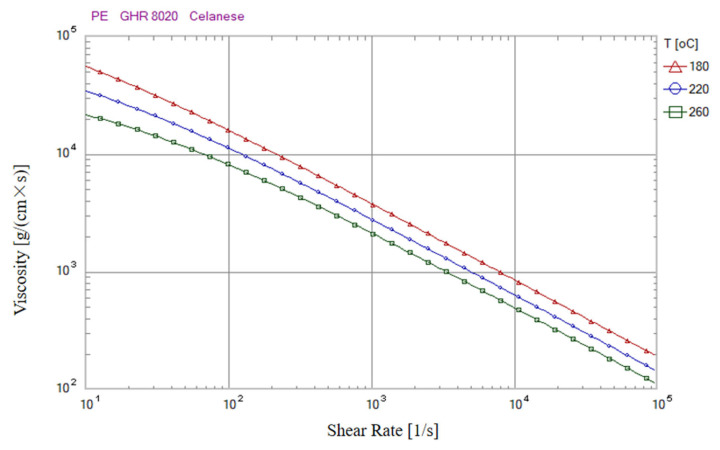
The viscosity curve of the original PE material for this study.

**Figure 10 polymers-14-00679-f010:**
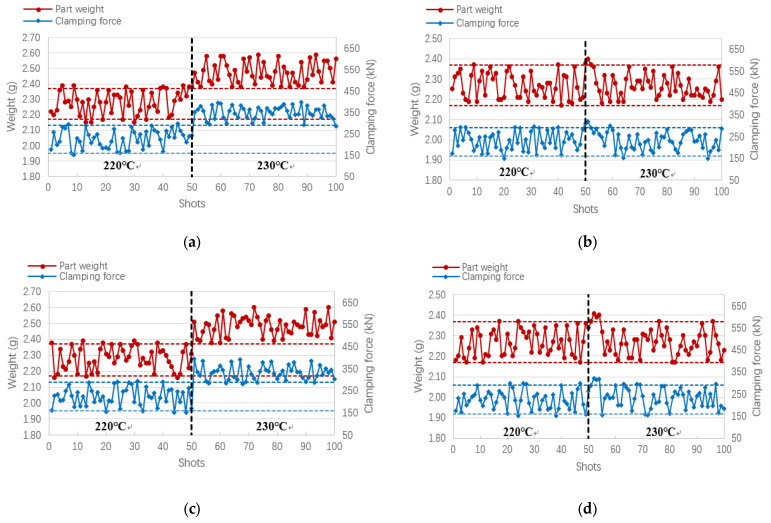
The change in clamping force and the weight of the product in 50 injection molding tests at a melt temperature of 220 °C and 230 °C, respectively. (**a**,**b**) original PE material without or with the quality monitoring method of this study; (**c**,**d**) primary recycled PE material without or with the quality monitoring method of this study; and (**e**,**f**) triple recycled PE material without or with quality monitoring method of this study.

**Table 1 polymers-14-00679-t001:** Pearson’s correlation coefficients (PCC) related to the correlation strength.

Range of |r|	Correlation Strength
0	No
0–0.25	Negligible
0.25–0.5	Poor
5–0.75	Moderate
0.75–1	Strong
1	Perfect

**Table 2 polymers-14-00679-t002:** Specifications of the injection molding machine of this study.

Machine Designation	Specification	Unit
Injection unit	Screw diameter (mm)	50
	Screw stroke (mm)	200
	Screw speed (mm/min)	300
	Injection pressure (bar)	2000
Clamping unit	Clamping force (kN)	1800

**Table 3 polymers-14-00679-t003:** Raw materials used in this study’s experiment.

Material (PE)	Recycling Number
Raw material	0
Once recycled material	1
Double recycled material	3

**Table 4 polymers-14-00679-t004:** The data of plastic bottle cap weight and corresponding key circumferential geometric dimensions under different injection molding parameters.

Parameters		Weight (g)	$\emptyset_{1}\;(\mathrm{mm})$	$\emptyset_{2}\;(\mathrm{mm})$	R (mm)
Margin position (mm)	18.5	2.31	1.72	1.22	0.96
	19.5	2.28	1.68	1.19	0.92
	20.5	2.23	1.63	1.16	0.89
Metering end point (mm)	94	2.24	1.64	1.17	0.90
	94.5	2.28	1.68	1.21	0.94
	95	2.32	1.72	1.25	0.98
Metering time (s)	7.5	2.22	1.63	1.15	0.91
	8	2.25	1.67	1.20	0.95
	8.5	2.29	1.70	1.23	0.98

**Table 5 polymers-14-00679-t005:** Important parameters related to the injection molding experiment in this study.

Item	Unit	Parameters
Melt temperature	°C	220
Mold temperature	°C	30
Cooling temperature	°C	30
Cooling time	s	20
Margin position	mm	18.5, 19.5, 20.5
Metering end point	mm	94.0, 94.5, 95.0
Metering time	s	7.5, 8, 8.5

**Table 6 polymers-14-00679-t006:** Important parameters related to the injection molding experiment in this study.

Group	Margin Position (mm)	Metering End Point (mm)	Metering Time (s)
1	18.5	94.5	8
2	19.5	94.5	8
3	20.5	94.5	8
4	19.5	94.0	8
5	19.5	94.5	8
6	19.5	95.0	8
7	19.5	94.5	7.5
8	19.5	94.5	8
9	19.5	94.5	8.5

**Table 7 polymers-14-00679-t007:** The data of clamping force and product weight for different molding parameters.

Parameters		Weight (g)	F (kN)
Margin position	18.5	2.31	226
	19.5	2.28	213
	20.5	2.23	186
Metering end point	94	2.24	185
	94.5	2.28	215
	95	2.32	228
Metering time	7.5	2.22	180
	8	2.25	210
	8.5	2.29	230

**Table 8 polymers-14-00679-t008:** Results of product weight of raw and recycled materials with and without quality monitoring methods in this study.

Item	Weight	Raw Material		Once-Recycled Material		Double-Recycled Material	
Shot number		1–50	51–100	1–50	51–100	1–50	51–100
Without quality monitoring	Average	2.28	2.47	2.28	2.48	2.28	2.50
	Range	0.24	0.21	0.23	0.22	0.24	0.22
	SD	0.0756	0.0695	0.0712	0.0608	0.0733	0.0691
With quality monitoring	Average	2.26	2.27	2.25	2.27	2.27	2.33
	Range	0.19	0.18	0.20	0.18	0.20	0.30
	SD	0.0415	0.0430	0.0463	0.0484	0.0615	0.0802

## Data Availability

Not applicable.

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
