# Peer review of "Intelligent Predicting of Product Quality of Injection Molding Recycled Materials Based on Tie-Bar Elongation"

_polymers, 2022, doi:10.3390/polym14040679_

Round 1
Reviewer 1 Report
Please see the attachment.

Author Response
Dear Sir,
Highest appreciate for your valuable comments, which are very useful and improvement on this project research way, please allow me to deliver thanksgiving again.
Best Regards,
Han-Jui Chang

Reviewer 2 Report
In this manuscript the quality prediction of injection molding of recycled material is addressed. The variation of clamping force is used as the main measurable parameter for this purpose. They show that the quality of the product falls in the standard range when they use their quality monitoring method.
The work is thorough and interesting and should be published. The results are clearly presented and can be exploited by researchers and industries to control the quality of the products of recycled materials with injection molding method.
There are a few comments that are not necessary to be addressed in order to have this work published here
- The Hooke’s law is only applicable when the strain is small and material is its linear behavior range. Do we know how big the tie bar deformations are? I assume it’s a small number but it might be good to be mentioned.
- The plot labels need to get fixed in Fig. 10.
Author Response
Dear Sir,
Highest appreciate for your valuable comments, which are very useful and improvement on this project research way, please allow me to deliver thankgsiving again.
Best Regards,
Han-Jui Chang
